# Anti-inflammatory role of APRIL by modulating regulatory B cells in antigen-induced arthritis

**Adriana Carvalho-Santos**[1,2]*, **Lia Rafaella Ballard Kuhnert**[2], **Michael Hahne**[3], **Rita Vasconcellos**[2], **Carla Eponina Carvalho-Pinto**[2], **Déa Maria Serra Villa-Verde**[1,4,5,6]*

**1** Laboratory on Thymus Research, Oswaldo Cruz Institute, Oswaldo Cruz Foundation, Rio de Janeiro, RJ, Brazil, **2** Experimental Pathology Laboratory, Department of Immunobiology, Biology Institute, Fluminense Federal University, Niterói, RJ, Brazil, **3** Institut de Génétique Moléculaire de Montpellier, Université de Montpellier, CNRS, Label "Equipe FRM", Montpellier, France, **4** National Institute of Science and Technology on Neuroimmunomodulation, Oswaldo Cruz Institute, Oswaldo Cruz Foundation, Rio de Janeiro, RJ, Brazil, **5** Rio de Janeiro Research Network on Neuroinflammation, Oswaldo Cruz Institute, Oswaldo Cruz Foundation, Rio de Janeiro, RJ, Brazil, **6** INOVA-IOC Network on Neuroimmunomodulation, Oswaldo Cruz Institute, Oswaldo Cruz Foundation, Rio de Janeiro, RJ, Brazil

* adcarvalhosantos@gmail.com (ACS); dvv@ioc.fiocruz.br, deavillaverde@gmail.com (DMSVV)

**Data Availability Statement:** All relevant data are within the manuscript.

**Funding:** This study was supported by grants from Conselho Nacional de Desenvolvimento Científico e

## Abstract

APRIL (A Proliferation-Inducing Ligand), a member of the TNF superfamily, was initially described for its ability to promote proliferation of tumor cells *in vitro*. Moreover, this cytokine has been related to the pathogenesis of different chronic inflammatory diseases, such as rheumatoid arthritis. This study aimed to evaluate the ability of APRIL in regulating B cell-mediated immune response in the antigen-induced arthritis (AIA) model in mice. AIA was induced in previously immunized APRIL-transgenic (Tg) mice and their littermates by administration of antigen (mBSA) into the knee joints. Different inflammatory cell populations in spleen and draining lymph nodes were analyzed using flow cytometry and the assay was performed in the acute and chronic phases of the disease, while cytokine levels were assessed by ELISA. In the acute AIA, APRIL-Tg mice developed a less severe condition and a smaller inflammatory infiltrate in articular tissues when compared with their littermates. We also observed that the total cellularity of draining lymph nodes was decreased in APRIL-Tg mice. Flow cytometry analysis revealed an increase of CD19+IgM+CD5+ cell population in draining lymph nodes and an increase of CD19+CD21hiCD23hi (B regulatory) cells in APRIL-Tg mice with arthritis as well as an increase of IL-10 and CXCL13 production *in vitro*.

## Introduction

Rheumatoid arthritis (RA) is a chronic inflammatory autoimmune disorder characterized by synovitis and proliferative changes in the synovial membrane. Immune complexes are formed in the joint and elicit an inflammatory response that leads to the destruction of the synovial lining [1]. The development and persistence of fully established rheumatoid synovitis requires

Tecnológico (CNPq) https://www.cnpq.br (DV-V Grants: 482028/2009-2 and 305927/2010-8); Fundação Carlos Chagas Filho de Amparo à Pesquisa do Estado do Rio de Janeiro (FAPERJ) – https://www.faperj.br (ACS Grant: E-26/102.500/ 2010). The study was also supported by a grant from MercoSur https://www.mercosur.int through the Fund for Structural Convergence (FOCEM); by intramural funds from Oswaldo Cruz Foundation (Fiocruz) www.fiocruz.br and from Fluminense Federal University (UFF) https://www.uff.br. There was no additional external funding received for this study. The funders had no role in study design, data collection and analysis, decision to publish, or preparation of the manuscript.

**Competing interests:** The authors have declared that no competing interests exist.

the collaboration of a variety of cells including synovial fibroblasts and synovial macrophages as the core target cells of RA, B and T cells, dendritic cells (DCs) and other cells of the immune system [1, 2]. It is important to point out that the activation of T cells in rheumatoid synovia appears to be strongly B-cell dependent [3], and in addition, the inflamed synovia is a site of significant B-cell infiltration, expansion, and terminal differentiation [4]. B cells have been linked to the pathogenesis of several autoimmune diseases based on their biological functions, such as antigen presentation, cytokine secretion and autoantibody production [5].

APRIL, as a homologous for B cell-activating factor (BAFF), causes the accumulation of plasma cells in the joint, further increasing the production of inflammatory cytokines such as TNF, IL-1 and IL-6 [6]. APRIL is one of the latest members cloned from the TNF superfamily, with a reported capacity to stimulate the proliferation of tumor cells *in vitro* [7]. Unlike most other TNF family members, APRIL is never expressed at the cell membrane. It is cleaved in the Golgi apparatus by a furin convertase to release a soluble active form, which is subsequently secreted [7–10]. Immune cell subsets such as monocytes, macrophages, dendritic cells, neutrophils and T-cells can express APRIL, whose activity takes place through interaction with transmembrane activator and calcium modulator cyclophilin ligand interactor (TACI) and B cell maturation antigen (BCMA) receptors [11, 12]. Furthermore, APRIL is also able to interact with heparin sulfate proteoglycans (HSPGs) [10, 13–15]. APRIL has been related to different autoimmune diseases including systemic lupus erythematosus (SLE); rheumatoid arthritis (RA) and diabetes [16–20].

Biological functions have been suggested like increase B-cell survival, co-stimulation of B-cell proliferation, antigen presentation in B-cell, in addition, APRIL has been found to promote immunoglobulin class switch recombination [7, 21–23]. Reports already described the potential of recombinant APRIL to act as a costimulator of primary B and T cells *in vitro*. Also, *in vivo* treatment was able to promote splenomegaly with B cell population expansion and an increase in the percentage of activated T cells [24]. It is important to note that in the pathogenesis of RA, B cells are thought to play a crucial function via the production of autoantibodies. In recent years, the existence of a B-cell subset with regulatory functions in the immune response to pathogens and autoantigens has been demonstrated. These regulatory cells have become a target for the study of autoimmune and infectious diseases [25]. Then, together with the reported capacity of APRIL to stimulate B cells *in vitro*, this molecule was suggested to be a strong disease promoter. A study conducted on peripheral blood B cells in patients with RA and healthy controls showed that RA patients have intact and transitional B cells significantly impaired as well as elevated APRIL serum levels, whereas no difference was found in terms of BAFF levels [1, 26]. We found that APRIL can be a negative regulator of inflammatory disease [18]. Another work recently showed that APRIL enhanced the proliferation and invasion capacity of fibroblast-like synoviocyte in the adjuvant-induced arthritis model [27].

The antigen-induced arthritis (AIA) model is a rodent model of RA, characterized by cellular infiltration of joints and periarticular tissue by neutrophils, macrophages, lymphocytes, and dendritic cells; pannus formation and destruction of cartilage and bone, all features resembling those of human RA [28–30]. In our study, we aimed to clarify the role of APRIL in AIA model using APRIL-Transgenic (APRIL-Tg) mice.

## Material and methods

### Mice

APRIL-Tg mice of C57BL/6 background were generated as previously described [18, 31]. They are heterozygous for the transgene and in all experiments performed, wild type littermates were used as controls. Mice were obtained from the Montpellier Institute of Molecular

Genetics (Montpellier, France) and maintained at the animal facility of the Oswaldo Cruz Foundation (Rio de Janeiro, Brazil) under specific conditions, at 21°C ± 2°C on a 12 h light/dark cycle. The care and handling procedures were in accordance with guidelines of the FIO-CRUZ Ethics Committee for the Use of Experimental Animals (CEUA LW-8/1- P-54/09).

### Antigen-induced arthritis (AIA)

Male APRIL-Tg mice and their control littermates, aged 8–10 weeks, were immunized subcutaneously (s.c.) with 100μg of methylated BSA (mBSA; Sigma-Aldrich, Saint Louis, USA, A-1009G), dissolved in 50μL of phosphate buffered saline (PBS; Sigma-Aldrich, Saint Louis, USA, P4417) and emulsified with equal volume of complete Freund adjuvant (CFA, Sigma-Aldrich, Saint Louis, USA, F5881), supplemented to 3mg/mL *Mycobacterium tuberculosis* (Difco, Glostrup, Denmark, H37Ra). Fourteen days after immunization, arthritis was induced by intraarticular inoculation of 10μg of mBSA in 10μL of sterile saline in the right knee joint (day 0), leading to development of severe acute synovitis associated with subsequent cartilage and bone erosion in the arthritis joints.

### Clinical assessment of AIA

The clinical evaluation was accompanied by two pathways during the course of AIA at definite time points. First, the swelling of the knee joint was recorded as the difference between the right (arthritic) and left (unaffected) knee diameter using a caliper rule (Vernier Caliper 150mm). The clinical severity of arthritis was graded as follows: 0, normal; 1, locomotion change; 2, limping and protecting paw (suspending the paw); 3, walking with bated paw. All clinical evaluations were performed in a blind manner [32–34].

### Histopathological examination

Knee joints were removed, fixed in 10% buffered formalin for 24 h and after decalcified in 10% EDTA (Sigma- Aldrich, Saint Louis, USA, E4884) for 4 weeks. The joints were subsequently embedded in paraffin, sectioned into 5μm thickness, and stained with hematoxylin and eosin (H&E). The specimens were evaluated by microscopy LEICA DM5500 B, (Leica Geosystems, Heerbrugg, Switzerland) using 10X and 20X objectives for magnification, according to the infiltration of cells in the articular cavity, inflammation in synovia and periarticular tissue, cartilage and bone erosion, synovial hyperplasia, and pannus formation.

### Total leukocytes count in lymphoid organs

The lymphoid organs, inguinal lymph nodes and spleen, were gently removed and macerated using a 40μm cell strainer (Falcon, Corning Incorporated, NY, USA, 352340) in RPMI 1640 medium (Sigma-Aldrich, Saint Louis, USA—R6504) with 10% Fetal Bovine Serum (FBS, Invitrogen Life Technology, Waltham, USA, 12657029) to obtain single-cell suspensions. They were then centrifuged at 1200 rpm/10 min. Lymph node pellets were resuspended in 2.0 mL of RPMI 10% FBS. The spleen pellets were first treated with 1.0 mL of Ammonium-Chloride-Potassium-ACK Lysing Buffer for 1 min, centrifuged at 1200 rpm/10 min and then also resuspended in 2.0 mL of RPMI 10% FBS. Counting was carried out in a Neubauer chamber at a dilution of 1:50, with trypan blue 10:10 v/v added.

### Phenotypical characterization by flow cytometric analysis

At day 3 after arthritis induction, single cell suspensions were prepared from inguinal lymph nodes and spleen. For surface staining, $10^6$ cells were incubated for 10 min with mouse IgG to

**Table 1. Labeled anti-mouse antibodies.**

| anti-mouse antibodies | Fluorochrome | References | dilution |
|---|---|---|---|
| CD4 | FITC/PE | BD | 1:100/1:100 |
| CD8 | APC/APCH7 | EBIOSCIENCE | 1:50/1:100 |
| CD19 | PeCy7 | SOUTHERN | 1:50 |
| CD5 | PE | BD | 1:50 |
| CD21 | APC | BD | 1:100 |
| CD23 | PE | BD | 1:100 |
| CD25 | APC | BD | 1:50 |
| FoxP3 | PE | EBIOSCIENCE | 1:50 |
| IgM | PE | BD | 1:100 |

block Fc receptors and then stained with fluorochrome coupled antibodies in FACS buffer, containing PBS; 2% FBS and 0,01% sodium azide (Sigma-Aldrich, Saint Louis, USA, S2002) for 20 min at 4˚C (Table 1). For gate strategy, in the region referring to individual cells (singlets), we excluded cells aggregated into doublets, combining the parameters of FSC-A (forward scatter area) x FSC-H (forward scatter height) [35, 36]. BD Accuri C6 cytometer (BD Bioscience, USA) was used for data acquisition, and the initial gate strategies depicted were applied (S1 Fig). The analyses were performed using the program Cflow (BD Bioscience, USA).

## Cytokine assays

Cell suspensions of inguinal lymph nodes and spleen were cultured ($10^6$ cells/mL) at 37˚C, 5% $CO_2$ in RPMI 1640 medium containing 10% FBS, 2nM sodium pyruvate, 10nM HEPES, 15 L-glutamine, 5μg/mL penicillin in the presence of 25 μg of mBSA or 2 μg of anti-CD3. Supernatants were harvested after 48 h of incubation and analyzed for levels of secreted cytokines using enzyme-linked immunosorbent assay (ELISA) procedures. Briefly, microplates (Thermo Scientific Nunc., Rochester, USA) were coated with capture anti-mouse antibody dissolved in PBS and incubated overnight at room temperature. After blocking with 300 μL of reagent diluent (1% BSA in PBS, pH 7.2–7.4), samples in duplicate and standards were added to the wells, followed by incubation for 2 h at room temperature. Finally, biotinylated anti-mouse antibody was added, that was detected by incubation with streptavidin-HRP plus substrate solution. Plates were washed at least three times between each step with 0.05% Tween® 20/PBS. OD was measured at 450nm. Samples of periarticular tissue extract and serum were also used to detect the cytokines. Briefly, 100 mg of tissue were homogenized in 1 mL of PBS (0.4M NaCl and 10 mM NaPO4) containing antiproteases (0.1 mM phenylmethylsulfonyl fluoride, 0.1 mM benzethonium chloride, 10 mM EDTA, and 20 kallikrein inhibitor units of aprotinin A) and 0.05% Tween® 20. The samples were then centrifuged for 10 min at 3,000g, and the supernatants were assessed by ELISA at a 1:3 dilution in PBS. Serum samples were used at 1:100 dilution. All samples were assayed in duplicate. All procedures were in accordance with instructions supplied by the manufacturer (Duoset Kits; R&D Systems, Minneapolis, MN: IFNγ, IL-10, CCL2, CXCL12 and CXCL13).

## Serum anti-mBSA antibody (Ab) levels

The presence of anti-mBSA Abs in the serum was determined by ELISA. Briefly microplates were coated with 5μL/mL mBSA dissolved in PBS and incubated overnight at 4˚C, blocked with 1% gelatin/PBS, and then incubated with serial dilutions of the testing sera. Anti-mouse-

HRP IgG, IgG1 and IgM (Southern Biotechnology) were used and detected by adding TMB/$H_2O_2$ solution. Plates were washed at least three times between each step with 0.05% Tween® 20/1% gelatin/PBS. OD was measured at 450nm.

## Statistical analysis

Statistical analysis was performed using the non-parametric Mann-Whitney U test, the two-way ANOVA followed by Bonferroni post-test and, the One-Way analysis of variance followed by Newman-Keuls tests. Data were expressed as mean ± SEM and *P* values less than 0.05 were considered significant.

## Results

### APRIL-Tg mice present less clinical and inflammatory responses to AIA

To better understand the role of APRIL in the AIA model, we used 8–10-week-old C57BL/6 APRIL-Tg mice and their control littermates, that were immunized with mBSA-CFA and challenged, after fourteen days, by an intraarticular injection of 10μg mBSA. Clinical signs of arthritis were evaluated by disease score on days 1, 2 and 3 post-induction, as described previously [32–34]. At day 3 after challenge, representing the acute phase of the disease, APRIL-Tg mice displayed moderate signs of arthritis (1.00 ± 0.00) when compared with their littermates (1.75 ± 0.11; **p < 0.01). The joint swelling (Δ mm between right and left knee) of APRIL-Tg mice group at the same time point showed a lower value (0.29 ± 0.07) than littermate group (0.60 ± 0.11), but statistically not significant (p > 0.05) (Fig 1A and 1B). These data show that APRIL-Tg mice presented more resistance to development of arthritis. Furthermore, APRIL-Tg mice showed lower cellularity in lymphoid organs. Inguinal lymph nodes and spleen were removed at day 3 and 10 after arthritis induction. At day 3 of AIA the total cellularity in inguinal lymph nodes of transgenic mice was lower in contrast to littermate mice (4.27 ± 0.47 and 8.27 ± 1.16, respectively, **p < 0.01). In spleen, no difference was observed between the groups upon arthritis induction (p = 0.29) (Fig 1D and 1E). At day 10, the cellularity of lymph nodes was decreased in APRIL-Tg mice (1.76 ± 0.28) when compared with their littermates (3.05 ± 0.35, p* = 0.02). The same profile was observed in the spleen of APRIL-Tg mice (19.24 ± 1.50) in relation to littermates (25.93 ± 1.56, p* = 0.02), (Fig 1F and 1G).

The AIA model develops a monoarticular inflammatory arthritis associated with antigen-specific immune response. The histopathological features of AIA include cellular infiltration of joints and periarticular tissue by inflammatory cells, and destruction of cartilage and bone with pannus formation. In our histopathological analysis, we did not observe inflammatory cells infiltrating in the articular cavity and adjacent tissues of APRIL-Tg mice (Fig 1K) when compared to littermates, in which an intense inflammatory infiltration was present at day 3 after arthritis induction (Fig 1J). The same profile was observed at day 10, chronic phase of disease (Fig 1L and 1M).

The evaluation of autoantibody production can be crucial to clarify the development of disease. Although APRIL-Tg mice have shown less severe disease, they maintained high anti-mBSA antibody levels in AIA. AIA APRIL-Tg mice presented increased levels of IgG (0.974 ± 0.01) and IgM (0.975 ± 0.11) in contrast to AIA littermates (IgG: 0.594 ± 0.07 and IgM: 0.467 ± 0.03; ***p<0.001 and *p<0.05 respectively) at day 3 of AIA (Fig 2A–2C). No difference was observed between AIA groups at day 10 (Fig 2B–2D). The elevated titers of anti-mBSA in APRIL-Tg mice with arthritis can be partially explained by the increased production of IgG1 subtype antibody when compared with their littermates (0.666 ± 0.02 and 0.516 ± 0.02, respectively, **p < 0.01) (Fig 2E).

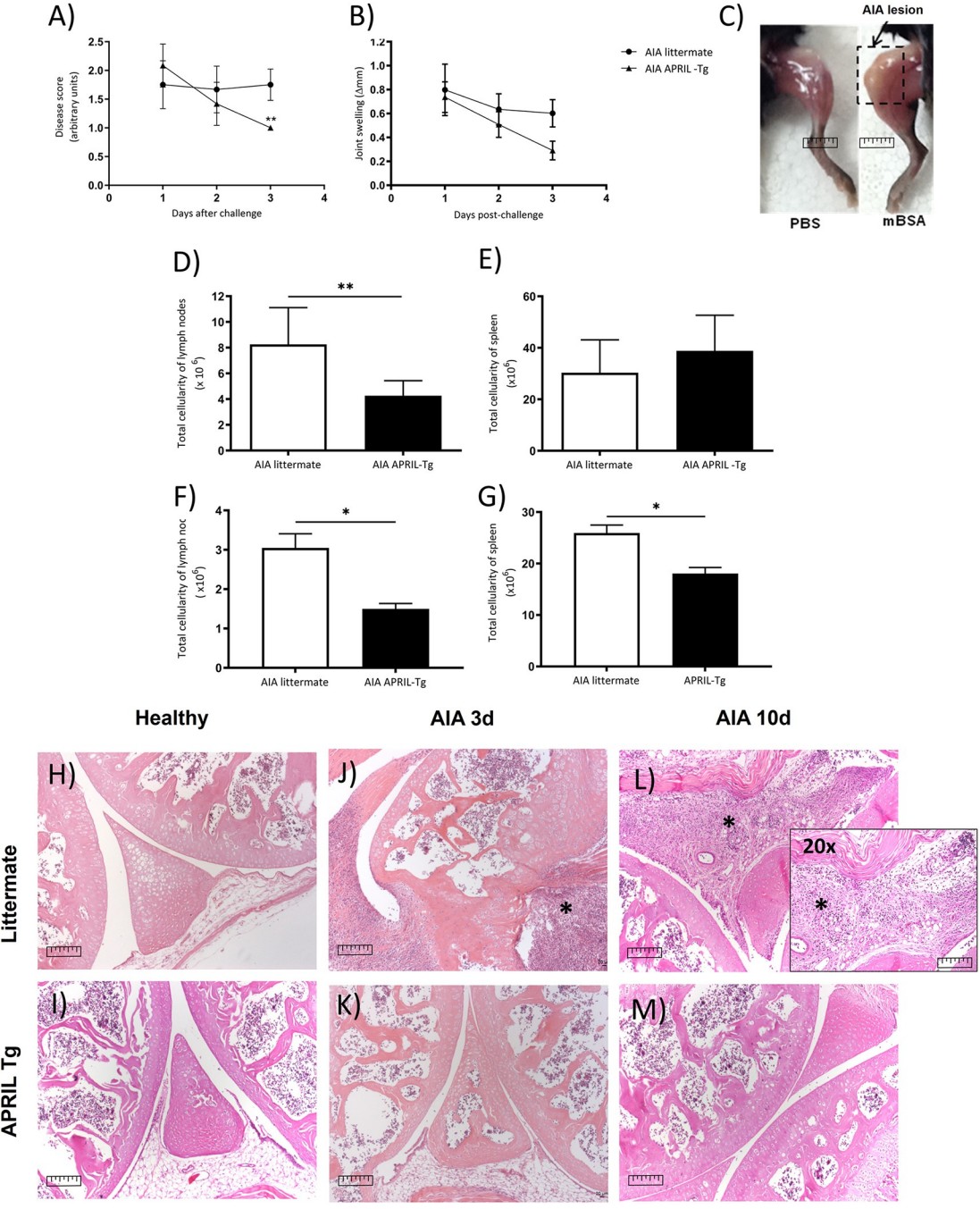

**Fig 1. C57BL/6 APRIL-Tg mice showed less inflammatory arthritis in acute and chronic phase.** A, APRIL-Tg mice displayed moderate signs of arthritis at day 3 post-challenge (1,00 ± 0.00; n = 6) when compared with their littermates (1.75 ± 0.11; n = 6; **p < 0.01). The disease score was described in Material and Methods. B, The joint swelling (Δ mm between right and left knee) of APRIL-Tg mice group at the same time point showed a lower value (0.29 ± 0.07, n = 8) than the littermate group (0.60 ± 0.11; n = 8), but statistically not significant (p > 0.05). C, Representative figure for AIA lesion. D, The total cellularity of inguinal lymph nodes of APRIL-Tg mice at day 3 of AIA was lower in contrast to littermate mice (D, **p < 0.01, n = 6). E, No difference was observed in the spleen between the groups upon arthritis induction (E, p = 0.29, n = 6). At day 10, lymph node and spleen cellularity were decreased in APRIL-Tg mice when compared with their littermates (F, G *p = 0.02, n = 4). Data are expressed as mean ± SEM and the statistical analysis was performed by two-way ANOVA followed by Bonferroni post-test and Mann-Whitney U test. Representative histology figure shows articular tissue stained with hematoxylin and eosin (H&E) of healthy mice (H, I), AIA APRIL-Tg mice (K, M) and AIA littermates (J, L) at days 3 and 10 post-antigenic challenge, respectively. Histopathological analysis, in magnification of 10X and 20X, revealed an intense mononuclear infiltrate in AIA littermates compared to APRIL-Tg mice. Bars correspond to 1 mm in figures and asterisks indicate hyperplasia of the synovial and pannus formation.

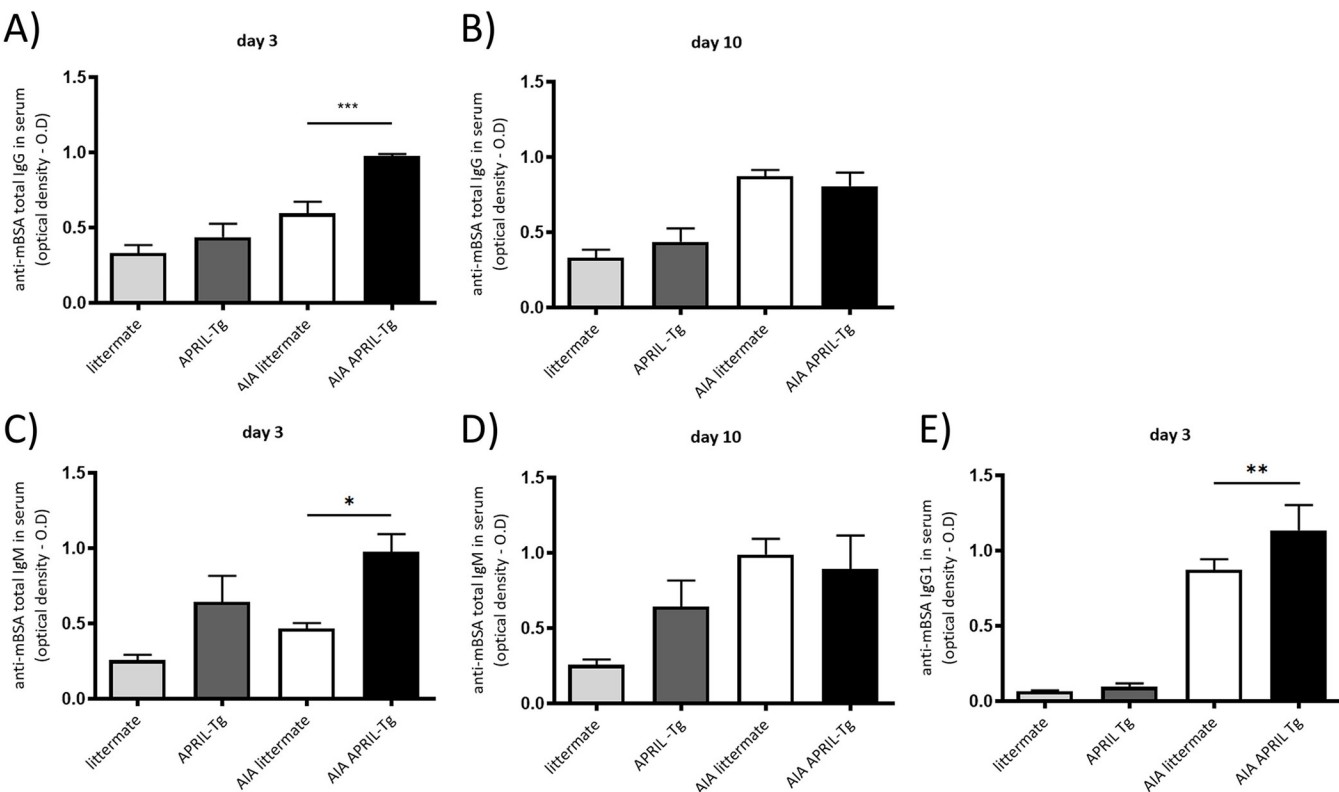

**Fig 2. C57BL/6 APRIL-Tg mice displayed high anti-mBSA antibody levels in AIA at days 3 and 10 after arthritis induction.** Anti-mBSA antibodies classes IgG, IgG1 and IgM were measured in serum by ELISA at days 3 and 10 after arthritis induction. AIA APRIL-Tg mice presented increased levels of IgG (A) and IgM (C) in contrast to AIA littermates at day 3, but no difference was observed between AIA groups at day 10 (B, D). In relation to IgG1subclass, the production was also higher in AIA APRIL-Tg mice than in AIA littermates at day 3 (E). Results are expressed as the mean ± SEM. Differences between groups were evaluated by One-Way analysis of variance followed by Newman-Keuls tests; *p < 0.05, **p < 0.01 and ***p<0.001.

## Overexpression of APRIL and the decrease of arthritis can be related to regulatory B cells

B cells are thought to play a crucial role in the pathogenesis of RA via the production of auto-antibodies. To date, there are no definitive markers that uniquely identify B regulatory cells (Bregs). In mice, the majority of reported Bregs express high levels of CD1d, CD24, and CD21 but variable levels of CD5 and CD23 [37]. The rheumatoid synovium is a site of significant B-cell infiltration, expansion, and differentiation [4]. This, together with the reported capacity of APRIL to stimulated B cells [17], led us to investigate B cell subset populations in lymphoid organs upon arthritis induction in APRIL-Tg mice. One of the important B regulatory subsets is the IgM$^+$CD5$^+$ B1 cells and, our data showed an increase of this phenotype in the spleen of AIA APRIL-Tg mice (16.60 ± 0.96; *p<0.05) in relation to AIA littermates (8.42 ± 2.44) and control groups, APRIL-Tg (9.83 ± 2.07) and littermates (6.58 ± 0.20) (Fig 3A and 3B). Similar results were observed in draining lymph nodes of AIA APRIL-Tg mice when compared with AIA littermates and control groups (data not shown).

Different approaches led to the identification that not all B cells exert suppressive function, but only those producing IL-10, that is an immunoregulatory cytokine [38]. The transitional 2 marginal zone precursor (T2-MZP) Bregs can represent an important population for its capacity to suppress autoimmune diseases like arthritis in mice [39]. These splenic cells can be identified by the CD19$^+$CD21$^{hi}$CD23$^{hi}$ phenotype. Moreover, the suppression of arthritis

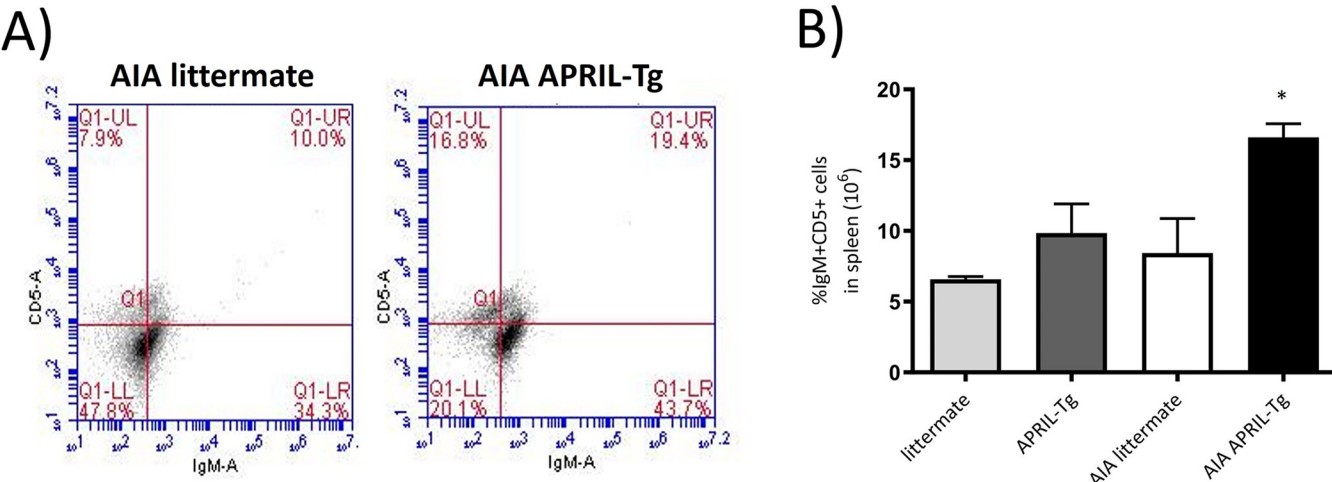

**Fig 3. Increase of B1 cells in spleen of C57BL/6 APRIL-Tg mice with AIA.** Spleens were removed at day 3 after intraarticular challenge with mBSA and flow cytometry technique was used to determine the frequency of CD5+IgM+ cells on CD19+ gated [35] in AIA and control mice (littermate and APRIL-Tg mice). A, Representative dot plot showing the frequency of B1 cell population in APRIL-Tg and littermate mice with AIA. B, Quantitative analysis revealed an increase of IgM+CD5+ cell population in AIA APRIL-Tg mice when compared with the other groups (AIA littermate, APRIL-Tg and littermate controls, n = 4 per group). Data are expressed as mean ± SEM and the statistical significance was determined using One-Way ANOVA followed by Newman-Keuls tests; *p < 0.05.

development seemed to be IL-10 production dependent [18, 39]. We showed that splenic B cells CD21$^{hi}$CD23$^{hi}$ gated on CD19 were more frequent in AIA APRIL-Tg mice (9.95 ± 0.61; *p<0.05) than in AIA littermates (8.15 ± 0.32), control APRIL-Tg mice (5.20 ± 0.05) and littermates (4.05 ± 1.45). The AIA littermate mice also had high numbers when compared to controls (**p< 0.01) (Fig 4A and 4B). Bearing in mind previous data showing that interactions between Bregs and T effector cells can result in the induction of both FoxP3+ Tregs and CD4+ T cells [38], lymph nodes were stained with anti-CD25 and anti-FoxP3 and analyzed by cytometry, as previously described. We showed an increase of CD4+CD25+FoxP3+ Treg cells in AIA APRIL-Tg mice (1.68 ± 0.22; *p<0.05) when compared with AIA littermates (1.15 ± 0.06) (Fig 5B). Nonetheless, no difference was observed in relation to CD4+ T cell frequency between AIA APRIL-Tg mice (29.64 ± 2.53) and AIA littermates (33.18 ± 4.86) (Fig 5A).

## Cytokine production in the context of APRIL and B regulatory cells in AIA model

B cells differentiating in the presence of inflammation or following priming by Th1 cells can produce high levels of proinflammatory cytokines. In addition, Bregs can exert their regulatory functions by secretion of cytokines, predominantly IL-10 [38, 40, 41]. The expression of CD5 and/or CD1$^{hi}$ identifies a rare splenic population of IL-10-producing B cells (B10 cells) that induce suppression in the contact hypersensitivity model [41]. Based on these data, we investigated the production of some cytokines and chemokines in lymphoid organs culture supernatant. Isolated cells of inguinal lymph nodes of APRIL-Tg and littermate mice with AIA at day 3 post-challenge were stimulated in microplates with mBSA or anti-CD3 as previously described. Upon mBSA stimulation, APRIL-Tg mice presented increased levels of IL-10 (884.1 ± 92.47; *p<0.05) and CXCL13 (650.3 ± 159.4) when compared with littermates (IL-10: 259.8 ± 190.4 and CXCL13: 278.2 ± 55.91, *p < 0.05) (Fig 6A–6D). We also observed an increase of IFNγ (3,027.0 ± 197.6, ***p < 0.001) and CXCL13 (482.8 ± 90.29, *p < 0.05) in APRIL-Tg mice with anti-CD3 stimulation (Fig 6B–6D). CXCL12 was decreased after mBSA

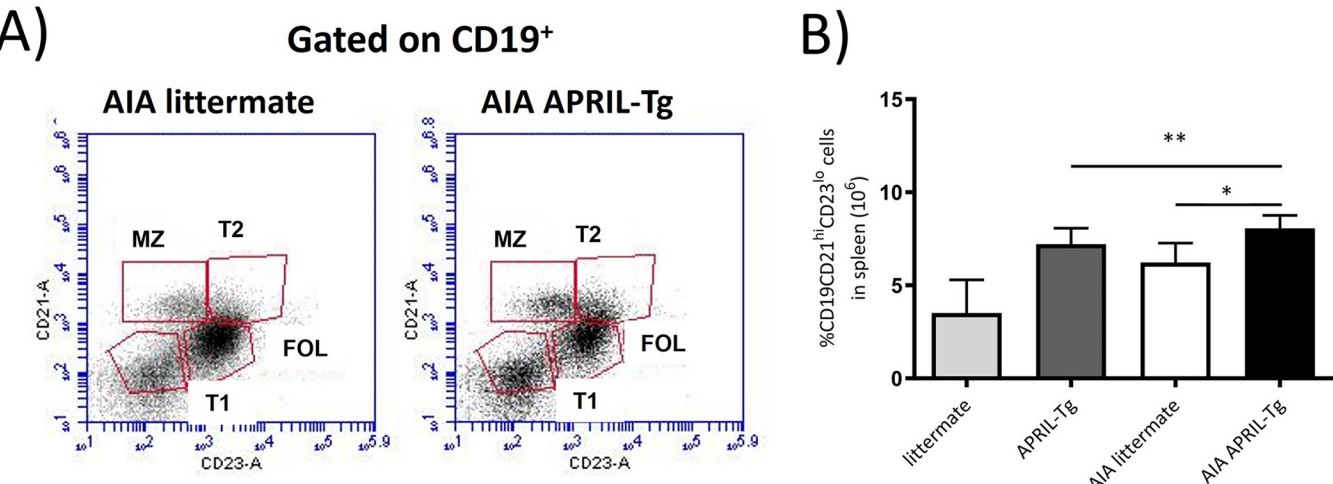

**Fig 4. Increased frequency of regulatory B cells (T2-MZP) in C57BL/6 APRIL-Tg mice with AIA.** Spleens were removed at day 3 after intraarticular challenge with mBSA and flow cytometry technique was used to determine the frequency of CD21^hi^CD23^hi^ B cells (T2-MZP) gated on CD19^+^ cells [36] in AIA and control mice (littermates and APRIL-Tg). A, Representative dot plot showed the frequency of T2-MZP cell population in APRIL-Tg and littermate mice with AIA. B, Quantitative analysis revealed an increase in the percentage of CD21^hi^CD23^hi^ B cells in AIA APRIL-Tg mice when compared with AIA littermates (*p < 0.05) and with APRIL-Tg and littermate controls (**p < 0.01, n = 4 per group). Data are expressed as mean ± SEM and the statistical significance was determined using One-Way ANOVA followed by Newman-Keuls tests.

and anti-CD3 stimulation in APRIL-Tg mice (570.0 ± 70.95 and 942.5 ± 221.5) when compared with littermates (1,583.0 ± 345.4 and 1,742.0 ± 189.2, respectively, *p < 0.05) (Fig 6C).

The maintenance of inflammatory response in arthritis can also be orchestrated by cytokine secretion at the articular tissue. The homogenate of tissues was performed as described at Material and Methods and cytokines were quantified by ELISA. No difference was observed between APRIL-Tg and littermate mice with AIA in relation to cytokines: IL-10, IFNγ and CCL2 (Fig 7A–7C). But increased levels of CXC12 and CXCL13 were detected in APRIL-Tg mice (4,267 ± 257.0 and 3,687 ± 39.69, respectively) when compared with littermates

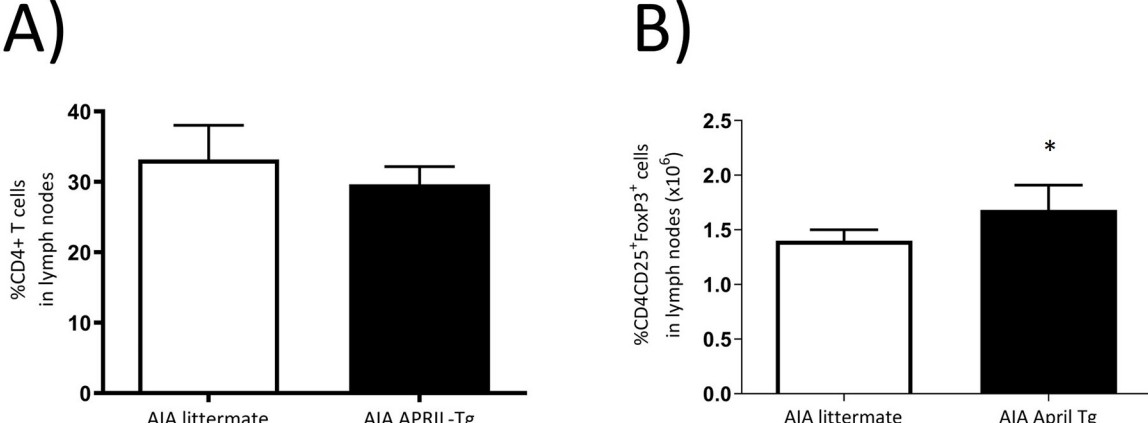

**Fig 5. Elevated number of CD4^+^CD25^+^FoxP3^+^ T regulatory cells in C57BL/6 APRIL-Tg mice with AIA.** Lymph nodes were removed at day 3 after intraarticular challenge with mBSA and flow cytometry technique was used to determine the frequency of CD4^+^ and CD4^+^CD25^+^FoxP3^+^ T cells in APRIL-Tg and littermate mice with AIA. No difference in the frequency of total CD4^+^ T cells was observed after induction of arthritis between the groups (A, n = 5 per group). However, the subpopulation of T regulatory cells, CD25^+^FoxP3^+^ cells gated on CD4 [36], was higher in AIA APRIL-Tg mice than AIA littermates (B, n = 7, *p < 0.05). Data are expressed as mean ± SEM and the statistical significance was determined using Mann-Whitney U test.

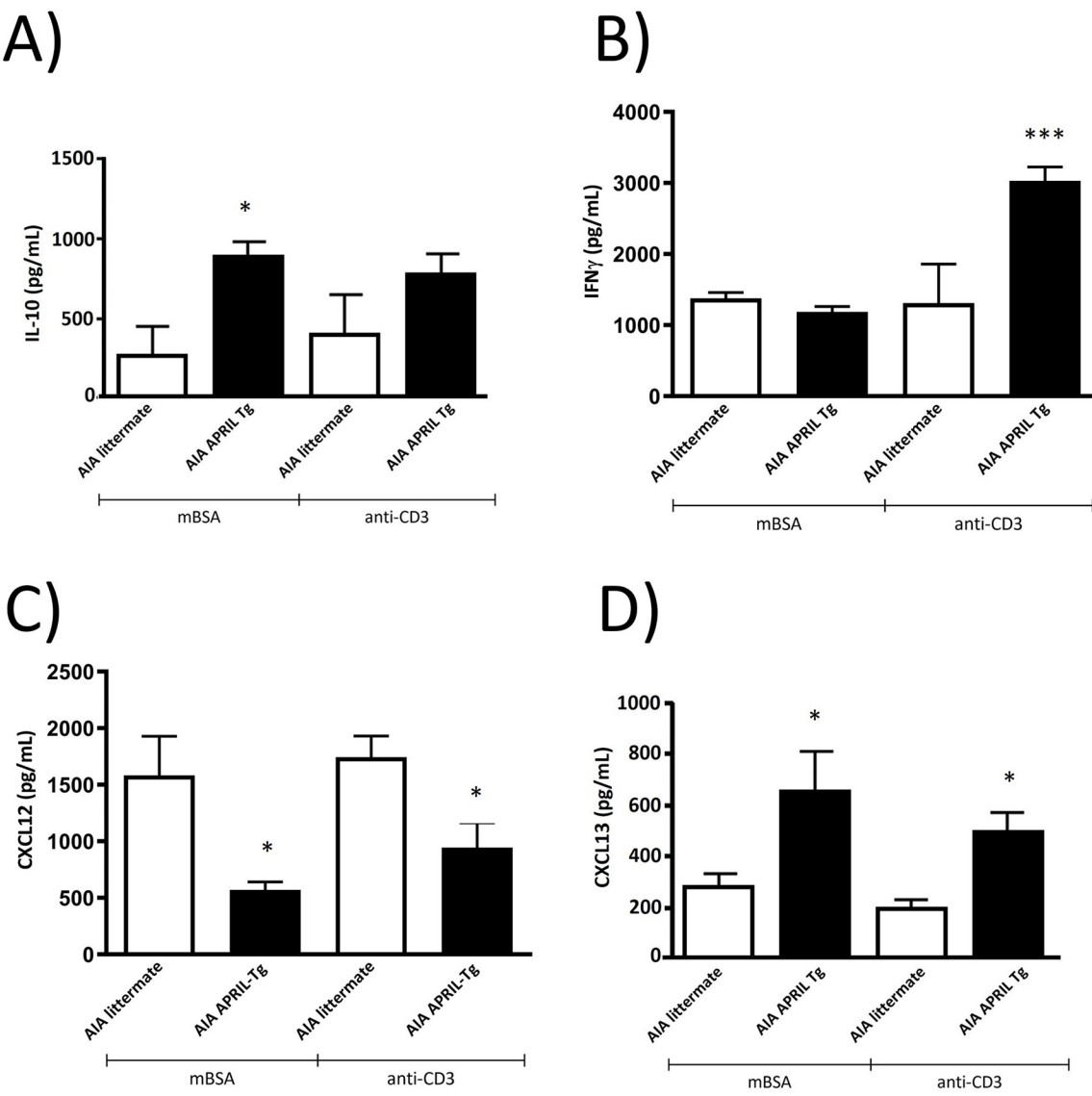

**Fig 6. Analysis of cytokine levels in supernatants of lymph nodes showed an altered profile in C57BL/6 APRIL-Tg mice with AIA.**
Inguinal lymph node cells of APRIL-Tg and littermate mice with AIA were isolated at day 3 post-challenge and stimulated *in vitro* with mBSA or anti-CD3, and cytokine levels were quantified by ELISA. Upon mBSA stimulation, APRIL-Tg mice presented increased levels of IL-10 (A) and CXCL13 (D) when compared with littermate mice (*p < 0.05). We also observed an increase of IFNγ (B, ***p < 0.001) and CXCL13 (D, *p < 0.05) in APRIL-Tg stimulated with anti-CD3 when compared to littermates. CXCL12 was decreased in contrast with littermate mice after mBSA and anti-CD3 stimulation (C, *p < 0.05). Data are expressed by mean ± SEM, n = 5 and, the statistical significance was determined using One-Way ANOVA followed by Newman-Keuls tests.

(2,345 ± 270.0 and 1,487 ± 342.1, respectively, **p < 0.01) (Fig 7D–7E). The chemokine CXCL13 binds to CXCR5 receptor and regulates homing of B cells whereas CXCL12 binds to CXCR4 receptor, acting as a potent chemoattractant for B cells, plasma cells, T cells and macrophages [42]. So, to investigate the secretion of these chemokines is very important to better understand the inflammatory tissue formation and the chronicity of the response. We also quantified CXCL12 and CXCL13 levels in the serum at the same time point of arthritis. AIA APRIL-Tg mice presented a decreased level of CXCL12 (39,456 ± 4,113) when compared with AIA littermate group (57,650 ± 5,522, *p < 0.05). No difference was detected in relation to CXCL13 (Fig 8A and 8B).

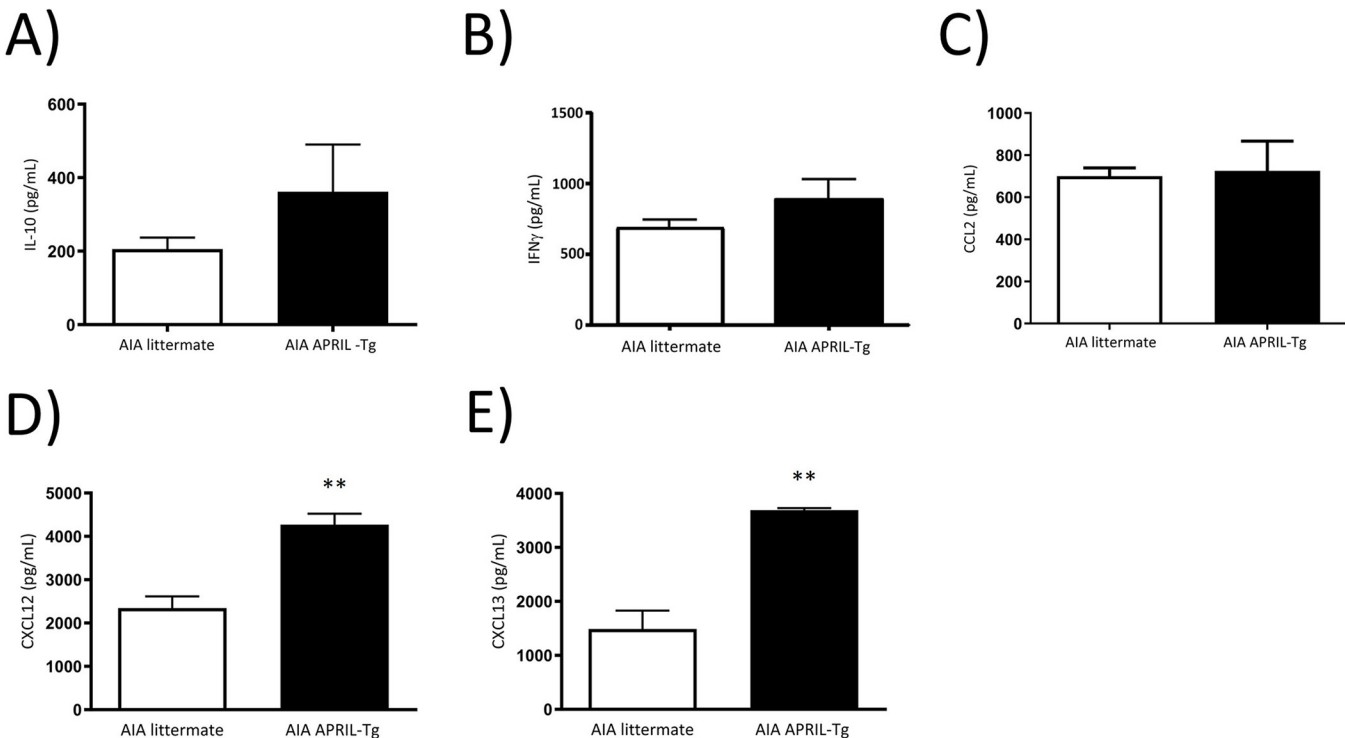

**Fig 7. Production of cytokines in periarticular tissue in C57BL/6 APRIL-Tg mice in the AIA model.** Periarticular tissues were removed at day 3 after induction of arthritis and the supernatants of tissue extracts were used to quantify cytokines by ELISA. No difference was observed between APRIL-Tg (n = 5) and littermate (n = 3) mice with AIA in relation to IL-10 (A), IFNγ (B) and CCL2 (C). But increased levels of CXCL12 (D) and CXCL13 (E) were detected in APRIL-Tg mice compared with littermates (**p < 0.01, n = 3 per group). Data are expressed by mean ± SEM and statistical analysis was performed using Mann-Whitney U test.

## Discussion

APRIL acts as a secreted factor and plays important roles in B cell biology [10]. The reported capacity of APRIL to stimulate tumor growth *in vitro* and regulate B cell functions raised the hypothesis that APRIL may be a disease promoter in autoimmune diseases such as RA [1, 7, 9, 18]. Previous reports described the ability of recombinant APRIL to act as a costimulator of primary B and T cells, and to stimulate IgM production by peripheral blood B cells *in vitro* [24], but its function is still not completely clear. The goal of this study was to investigate the role of APRIL as a regulator of B cell-mediated inflammatory arthritis using the AIA model in C57BL/6 APRIL-Tg mice and their littermates. Mice were preimmunized subcutaneously with mBSA and fourteen days after, arthritis was induced by intraarticular injection of the same antigen in knee joint. C57BL/6 APRIL-Tg mice with arthritis induced by mBSA antigen displayed lower clinical response when compared with littermates, which corroborate our previous data using the collagen-induced arthritis model (CIA) in DBA/1 mice. In that article we showed that decreased arthritis in DBA/1 APRIL-Tg mice was related to impaired anti-collagen type 2 (CII) antibody responses, considering IgG1 and IgG2a [18]. This is distinct to our present observations in AIA showing that C57BL/6 APRIL-Tg mice develop less severe disease but maintained high levels of anti-mBSA IgG. The latter can be partially explained by the increased production of IgG1 subtype antibody when compared with their littermates. Additional experiments evaluating other IgG subclasses are needed to better understand these results. Previous data have shown that APRIL transgenic mice have a normal development of B cells without any indication of hyperplasia and that the effect of APRIL on the humoral

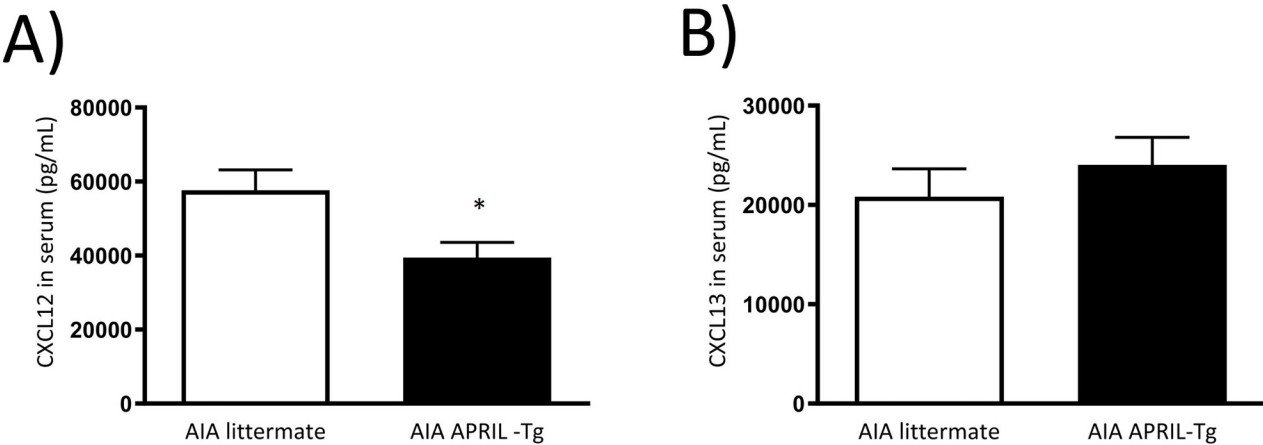

**Fig 8. Detection of the chemokines CXCL12 and CXCL13 in the serum of C57BL/6 APRIL-Tg mice with AIA by ELISA.** Serum samples were obtained at day 3 after induction of arthritis. APRIL-Tg mice (n = 9) presented increased levels of CXCL12 (A) compared to the littermate group (n = 6), *p < 0.05. No difference was observed in relation to CXCL13 (B). Data are expressed by mean ± SEM and the statistical significance was determined using One-Way ANOVA followed by Newman-Keuls tests.

immune response was associated with an increase in serum IgM level without altered IgG levels [1, 21].

B cell differentiation in the presence of inflammation following priming by Th1 cells can produce high levels of proinflammatory cytokines. Also, B regulatory cells exert their functions both by secretion of cytokines, predominantly IL-10, and by direct cell-cell contact, in which CD80 and CD86 play a pivotal role. In RA, the important cellular targets of Breg cell suppressions include CD4+ T cells, monocytes, and iNKT cells [40, 41, 43, 44]. About Breg cells, Fillatreau and collaborators in 2002 [45] described IL–10-producing B cells capable of diminishing the clinical manifestations in EAE. However, evidence supports the existence of additional regulatory mechanisms, besides IL-10 production, both in mice and humans, such as cell death induction and transforming growth factor β (TGF-β) and immunoglobulin M (IgM) production, among others [25, 45]. BAFF is a critical B-cell survival and differentiation factor and helps shape the innate marginal zone (MZ) B-cell pool by contributing to selection during normal B-cell ontogeny. APRIL is a BAFF analogue and despite having similarities, BAFF and APRIL have different functions in B-cell immunity [44, 46–48]. In mice, Breg cells express predominantly high levels of CD1d, CD24, and CD21 but variable levels of CD5, CD10, and CD23 [38, 49]. Studies in humans show that high levels of CD1d and CD24 are often associated with IL-10-production by B cells [50, 51]. In the CIA model in DBA/1 mice, an important suppressive role of naive T2-MZP B cells was shown, preventing the induction of arthritis. And this role of B regs has been associated with the production of IL-10 [39, 52].

Moreover, T cell activation in rheumatoid synovia appears to be strongly dependent on B cells [3, 18]. In addition, the inflamed articular tissue is a site of significant B cell infiltration, expansion, and differentiation [4]. Our data suggest that overexpression of APRIL and decrease of arthritis can be related to regulatory B cells. Flow cytometry analysis of inguinal lymph nodes and spleen of AIA C57BL/6 APRIL-Tg mice showed a higher percentage of IgM+CD5+ B1 cells when compared with AIA littermates at the same time of the clinical and pathological evaluation. It is important to note that this increase was above that shown by C57BL/6 APRIL-Tg mice without arthritis. Our recent work showed that DBA/1 APRIL-Tg mice with CIA displayed increased percentages of CD5low B1 cells in the spleen and peritoneum [18]. Original description of C57BL/6 APRIL Tg mice described an unaltered

homeostasis of spleen and peripheral lymph nodes when compared to control mice. In addition, transgenic mice display normal B cell development in spleen and no difference in the percentage of T1, T2, mature, or marginal B cells was observed comparing to littermate mice [21, 53, 54].

To evaluate the role of T2-MZP Breg cells in the suppression of inflammatory response after arthritis induction in C57BL/6 APRIL-Tg mice, we analyzed CD19[+]CD21[hi]CD23[hi] spleen cells. We observed that C57BL/6 APRIL-Tg mice with arthritis showed a higher percentage of T2-MZP cells, but, corroborating recent data, it is important to define this population as IL-10-producing B cells. A recent study showed that APRIL upregulates the expression of Breg markers by blood precursor-like MZ from HIV-uninfected individuals [48]. It had already been described in humans that a B cell subset which, in response to CD40 engagement, suppresses the differentiation of Th1 cells, and that this inhibitory effect is IL-10, but not TGFβ dependent, and requires signaling via CD80 and CD86 [41, 50]. IL-10 producing B cells in humans are mainly, although not exclusively, contained within the CD19[+]CD24[hi]CD38[hi] immature B cell subset which is equivalent to transitional B cells in mice [55]. Human CD19[+]CD24[hi]CD38[hi] Bregs also express high levels of CD1d and are CD5[+], suggesting that, at least in humans there may be some concordance between transitional Bregs and B10 cells previously described in experimental models [50]. Interactions between Bregs and T effector cells can result in the induction of both FoxP3[+] T cells and regulatory CD4[+] T cells producing IL-10 (Tr1). The defective function of Tregs in patients with active RA suggests that an unbalanced Treg versus T helper 1 (Th1) and Th17 response may favor the pathogenesis of the disease [56]. Corroborating previous data, our study showed that draining lymph nodes presented an increase of CD4[+]CD25[+]FoxP3[+] Treg cells in AIA APRIL-Tg mice. These results lead us to investigate a possible axis of regulating the immune system, involving the cytokine APRIL, regulatory B cells and anti-inflammatory cytokines. Irmler and collaborators in 2007 showed that IFNγ[-/-] AIA mice presented an increased inflammatory response, and treatment with the recombinant murine IFNγ into the knee joint during arthritis inducing was capable to attenuate the joint injury [29]. Another work showed that the inhibition of chemokine receptors CXCR1 and CXCR2 led to decreased neutrophil recruitment and cytokine production in AIA mice [57].

Recruitment, clonal selection, and expansion of B cells require a specialized milieu of secondary lymphoid organ chemokines and cytokines. Moreover, chemokines such as CCL19, CCL21, and CXCL13 are known to influence migration of B cells. CXCL13 binds the CXCR5 receptor and regulates homing of B cells and, recently it has been suggested as a prognostic marker for multiple sclerosis (MS), another chronic inflammatory autoimmune disease. Elevated CXCL13 levels were found in the cerebrospinal fluid (CSF) of patients with MS, neuroborreliosis and other inflammatory neurological diseases [42]. Also, a novel therapeutic antibody targeting CXCL13-mediated signaling pathway has been postulated for the treatment of autoimmune disorders [58]. Other important chemoattractant for B cells is the CXCL12 that binds to CXCR4 receptor [42, 59, 60]. Using ELISA, we observed in lymphoid organs culture supernatants upon mBSA stimulation, that AIA C57BL/6 APRIL-Tg mice presented increased levels of IL-10 and CXCL13 when compared with AIA littermates. In contrast CXCL12 was decreased after the same stimulus in AIA APRIL-Tg. Interestingly, APRIL pretreatment *in vitro* enhanced the production of inflammatory factors (IL-1α, IL-1β, IL-13, MCP-1 and TNF-α) by fibroblast-like synoviocytes (FLS) and decreased the production of IL-10, suggesting that APRIL may act as a mediator for facilitating the function of FLS [27]. In our study we observed increased levels of CXCL12 and CXCL13 in AIA C57BL/6 APRIL-Tg mice in the homogenate of articular tissue. Thus, the possible effective function of APRIL in B regulatory cell migration to the articular tissue needs to be better investigated. We also

quantified serum cytokines at the same time point of arthritis and observed that AIA APRIL-Tg mice presented a decreased level of CXCL12 and no difference in relation to CXCL13. Based on those data, the maintenance of inflammatory response in arthritis can also be orchestrated by secretion of cytokines at different microenvironments. Our data suggest that APRIL can be a predictor of resistance to antigen-induced arthritis in C57BL/6 APRIL-Tg mice and this effective role may be occurring through Breg cells that could reach the inflamed synovia and build an anti-inflammatory environment in arthritis. Recent studies described that APRIL and BAFF genes were upregulated in spinal-cord injury associated to autoimmunity [61]. Moreover, discussions have been held regarding the role of APRIL in human cancer development and autoimmune diseases, pointing this cytokine as an important therapeutic target [62, 63]. Together with the anti-inflammatory role of APRIL in chronic processes already demonstrated in the literature, our data suggest that APRIL has an anti-inflammatory function in the acute model of AIA.

## Supporting information

**S1 Fig. Gate strategy for analyzing lymphocytes subpopulations in the spleen and lymph nodes.** A, Spleen, and lymph node cells, obtained from littermate control mice as described in Material and Methods, were acquired on a BD Accuri C6 cytometer (BD Bioscience, USA). Dot plots representing the gate strategy for the lymphocyte total population region in the FSC (size) and SSC (complexity) parameters. B, Dot plot representing the gate strategy in the region referring to individual cells (singlets), excluding cells aggregated into doublets, combining the parameters of FSC-A (forward scatter area) x FSC-H (forward scatter height). After applying the initial gate strategies depicted, lymphocyte subsets were analyzed using the program Cflow (BD Bioscience, USA) as following, CD19+CD5+ B1 cells as referred in [35]; CD19 +CD21hiCD23hiCD24+T2-MZP B regs and CD4+CD25+FOXP3+ T regs as referred in [36]. (TIF)

## Author Contributions

**Conceptualization:** Adriana Carvalho-Santos, Rita Vasconcellos, Carla Eponina Carvalho-Pinto, Déa Maria Serra Villa-Verde.

**Data curation:** Adriana Carvalho-Santos.

**Formal analysis:** Adriana Carvalho-Santos, Rita Vasconcellos, Carla Eponina Carvalho-Pinto.

**Funding acquisition:** Adriana Carvalho-Santos, Déa Maria Serra Villa-Verde.

**Investigation:** Adriana Carvalho-Santos, Rita Vasconcellos, Carla Eponina Carvalho-Pinto, Déa Maria Serra Villa-Verde.

**Methodology:** Adriana Carvalho-Santos, Rita Vasconcellos.

**Project administration:** Carla Eponina Carvalho-Pinto, Déa Maria Serra Villa-Verde.

**Resources:** Adriana Carvalho-Santos, Michael Hahne, Rita Vasconcellos, Carla Eponina Carvalho-Pinto, Déa Maria Serra Villa-Verde.

**Supervision:** Adriana Carvalho-Santos, Rita Vasconcellos.

**Validation:** Adriana Carvalho-Santos, Rita Vasconcellos.

**Visualization:** Adriana Carvalho-Santos, Lia Rafaella Ballard Kuhnert, Carla Eponina Carvalho-Pinto.

**Writing – original draft:** Adriana Carvalho-Santos, Rita Vasconcellos.

**Writing – review & editing:** Adriana Carvalho-Santos, Lia Rafaella Ballard Kuhnert, Michael Hahne, Rita Vasconcellos, Carla Eponina Carvalho-Pinto, Déa Maria Serra Villa-Verde.

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
