## [Decision Letter · Decision Letter 0]

15 Dec 2023

PONE-D-23-29381Anti-inflammatory role of APRIL by modulating regulatory B cells in antigen-induced arthritisPLOS ONE

Dear Dr. Villa-Verde,

Thank you for submitting your manuscript to PLOS ONE. After careful consideration, we feel that it has merit but does not fully meet PLOS ONE’s publication criteria as it currently stands. Therefore, we invite you to submit a revised version of the manuscript that addresses the points raised during the review process.

We look forward to receiving your revised manuscript.

Kind regards,

Sadiq Umar

Academic Editor

PLOS ONE

Journal Requirements:

"This study was supported by grants from Conselho Nacional de Desenvolvimento Científico e Tecnológico (CNPq) https://www.cnpq.br  (DV-V Grants: 482028/2009-2 and 305927/2010-8); Fundação Carlos Chagas Filho de Amparo à Pesquisa do Estado do Rio de Janeiro (FAPERJ) – https://www.faperj.br (ACS Grant: E-26/102.500/2010). The study was also supported by intramural funds from Oswaldo Cruz Foundation (Fiocruz) www.fiocruz.br and Fluminense Federal University (UFF) https://www.uff.br."

"This work was supported by the Oswaldo Cruz Foundation intramural funding, as well as CNPq, and FAPERJ (Brazil). The work was also partially funded by a grant from MercoSur through the Fund for Structural Convergence (FOCEM). It was developed in the framework of the National Institute of Science and Technology on Neuroimmunomodulation (INCT-NIM; CNPq), Rio de Janeiro Research Network on Neuroinflammation (Faperj), INOVA-IOC Network on Neuroimmunomodulation (IOC/Fiocruz) and Experimental Pathology Laboratory of Fluminense Federal University."

"This study was supported by grants from Conselho Nacional de Desenvolvimento Científico e Tecnológico (CNPq) https://www.cnpq.br  (DV-V Grants: 482028/2009-2 and 305927/2010-8); Fundação Carlos Chagas Filho de Amparo à Pesquisa do Estado do Rio de Janeiro (FAPERJ) – https://www.faperj.br (ACS Grant: E-26/102.500/2010). The study was also supported by intramural funds from Oswaldo Cruz Foundation (Fiocruz) www.fiocruz.br and Fluminense Federal University (UFF) https://www.uff.br. The funders had no role in study design, data collection and analysis, decision to publish, or preparation of the manuscript."

Reviewers' comments:

Reviewer's Responses to Questions

**Comments to the Author**

1. Is the manuscript technically sound, and do the data support the conclusions?

Reviewer #1: Yes

Reviewer #2: Yes

2. Has the statistical analysis been performed appropriately and rigorously? 

Reviewer #1: Yes

Reviewer #2: Yes

3. Have the authors made all data underlying the findings in their manuscript fully available?

Reviewer #1: Yes

Reviewer #2: Yes

4. Is the manuscript presented in an intelligible fashion and written in standard English?

Reviewer #1: Yes

Reviewer #2: Yes

5. Review Comments to the Author

Reviewer #1: The manuscript has explained the anti-inflammatory effect of APRIL (A Proliferation-Inducing Ligand) in regulating B cell-mediated immune response in rheumatoid arthritis. Since inflammatory factors such as TNF-α has changed by APRIL, how did P38 MAPKs change in this experiment?

Briefly, the manuscript presented all the parameters in the questions above.

The text in figures is unclear and it is hard to read. I would appreciate it if you modified it. Thank you very much. Best regards,

Reviewer #2: The manuscript titled "Anti-inflammatory role of APRIL by modulating regulatory B cells in antigen-induced arthritis" explores APRIL's role in regulating the B cell-mediated inflammatory response in an antigen-induced arthritis mouse model. The study utilizes an mBSA-induced arthritis model, uncovering that APRIL overexpression correlates with reduced joint swelling, disease scores, and inflammatory infiltration in knee joint tissues. The study is focused on APRIL determined behavior of immune cells (mostly B cells) in mouse arthritis model.

While the anti-inflammatory role of APRIL has been documented in other contexts, its function under specific inflammatory conditions, such as arthritis, necessitates further examination. This study provides new insights into APRIL's anti-inflammatory effects in a mouse arthritis model. However, before publication, improvements are required. The following are my major and minor concerns:

Major Concerns:

- The study demonstrates differences in specific B cell and CD4+ T cell subpopulations, yet it remains unclear whether the total number of various immune cell types (T cells (CD4+, CD8+), B cells, myeloid cells (neutrophils, macrophages, dendritic cells, etc.)) is impacted by APRIL overexpression. A comparison of these numbers in the joint area, lymph nodes, and spleen is necessary

- All flow cytometry data should include a complete gating strategy. If necessary, part of this data can be placed in supplementary material to avoid overloading the figures

- In the histopathological comparison, was morphometry employed? How were conclusions regarding the inflammatory infiltrate drawn? Were there observable differences in other morphological changes between control and APRIL-tg groups? If so, these should be presented with corresponding morphometry

- The discrepancy in IgG titer is mostly attributed to the IgG1 subclass (0.974 (APRIL-Tg mice) vs 0.594 (ctrl mice)). However, the actual IgG1 titer shows a less significant difference (0.666 (APRIL-Tg mice) vs 0.516 (ctrl mice). How do authors explain this? The reasons for this and the titers of other subclasses should be addressed

- The study's conclusion lacks clarity regarding the precise role of APRIL-driven mechanisms linked to B cell behavior in arthritis. It is essential to elucidate why B cells, in particular, are the focus over other cell types in the progression of arthritis in the context of APRIL's effects (in fact APRIL affects many cell types). Additionally, the rationale for emphasizing B cells should be made explicit, considering their role in comparison to other immune cells within the APRIL-mediated response in arthritis

Minor Concerns:

- All plots should demonstrate individual points

- In Figure 2, cellularity in lymph nodes and spleen is compared, but the methodology is unclear. If flow cytometry was used, how was it normalized? This information is needed

- The study should address whether there is a gender dependency in APRIL expression and justify the exclusive use of male mice

- Details on single-cell suspension preparation should be added to the "Materials and Methods" section

- The image (histological slides) acquisition technique needs description and inclusion in the "Materials and Methods" section (microscope, magnification etc.)

- The current total of 11 figures could be consolidated into 5-6 for ease of reading, combining figures where feasible

- Figure 3 requires redesigning for clarity, particularly in labeling images A and D, which currently reference healthy mice but appear to indicate acute and chronic arthritis states in the figure

6. PLOS authors have the option to publish the peer review history of their article (what does this mean?). If published, this will include your full peer review and any attached files.

Reviewer #1: **Yes: **Mina Bagheri Varzaneh

Reviewer #2: No

---

## [Author Response · Author response to Decision Letter 0]

1 Apr 2024

Response to Reviewers

Reviewer #1: The manuscript has explained the anti-inflammatory effect of APRIL (A Proliferation-Inducing Ligand) in regulating B cell-mediated immune response in rheumatoid arthritis. 

Since inflammatory factors such as TNF-α has changed by APRIL, how did P38 MAPKs change in this experiment? 

Previous studies, using mouse macrophages associated the expression of APRIL with IL4 and TGFβ stimulation, and showed that the inhibition of p38MAPK abolished the increase of APRIL expression induced by both cytokines through different pathways (Jang YS, et al., 2009, Cytokine. 47: 43-47 doi:10.1016/j.cyto.2009.04.005; Jang YS et al., 2011. Molecules and Cells. 32: 251-255, doi:10.1007/s10059-011-1040-4). 

In addition, our group showed that APRIL is expressed via p38MAPK in a human breast cancer cell line (García-Castro A, et al., 2015, Carcinogenesis. 36: 574–584. doi:10.1093/carcin/bgv020). In another study, the authors evaluated role of p38 MAPK inhibitors in the control of rheumatoid arthritis, being an important candidate in therapy (Westra J and Limburg PC. p38 mitogen-activated protein kinase (MAPK) in rheumatoid arthritis. Mini Rev Med Chem. 2006 6:867-74, doi: 10.2174/138955706777934982. PMID: 16918493). 

However, the main objective of our work was to study the role of the cytokine APRIL based on the behavior of regulatory B cells. And one of the regulatory pathways for these cells is related to the anti-inflammatory function of the cytokine IL-10, which in turn can drive the differentiation process of regulatory T cells. These findings have been extensively discussed by Mauri C and collaborators (ref.[38] cited in the manuscript). In our study, regulatory T cells were more frequent in animals that overexpressed APRIL. Although these are very important data as they corroborate other findings, other tests still need to be carried out for better elucidation this point, as well as evaluating the production of TNF in this arthritis model. 

Briefly, the manuscript presented all the parameters in the questions above. 

The text in figures is unclear and it is hard to read. I would appreciate it if you modified it. 

We thank the Reviewer for this suggestion, the text in figures was modified for clarity of reading. 

Thank you very much. Best regards,

Reviewer #2: The manuscript titled "Anti-inflammatory role of APRIL by modulating regulatory B cells in antigen induced arthritis" explores APRIL's role in regulating the B cell-mediated inflammatory response in an antigen-induced arthritis mouse model. The study utilizes an mBSA-induced arthritis model, uncovering that APRIL overexpression correlates with reduced joint swelling, disease scores, and inflammatory infiltration in knee joint tissues. The study is focused on APRIL determined behavior of immune cells (mostly B cells) in mouse arthritis model. While the anti-inflammatory role of APRIL has been documented in other contexts, its function under specific inflammatory conditions, such as arthritis, necessitates further examination. This study provides new insights into APRIL's anti-inflammatory effects in a mouse arthritis model. However, before publication, improvements are required. The following are my major and minor concerns:

Major Concerns: 

- The study demonstrates differences in specific B cell and CD4+ T cell subpopulations, yet it remains unclear whether the total number of various immune cell types (T cells (CD4+, CD8+), B cells, myeloid cells (neutrophils, macrophages, dendritic cells, etc.)) is impacted by APRIL overexpression. A comparison of these numbers in the joint area, lymph nodes, and spleen is necessary. 

We thank the reviewer for this suggestion, in fact the evaluation of myeloid cells was not done in this article. Unfortunately, the images did not allow a precise definition of cell types. The comparison of different cell types in the joint area is an interesting point and merits new studies but it was not performed because of the difficulty in obtaining a good quantity of cells for cytometric evaluation. However, in our histopathological studies, we did not observe inflammatory cells infiltrating the articular cavity and adjacent tissues of APRIL-Tg mice when compared to littermates. This information was included in the “Results” section of the manuscript. The focus of the present work was to evaluate APRIL protection in AIA and the role of B cells, but new studies are interesting to extend our understanding of this issue. 

- All flow cytometry data should include a complete gating strategy. If necessary, part of this data can be placed in supplementary material to avoid overloading the figures 

We thank the Reviewer for this suggestion. The gating strategy was included in “Material and Methods” section of the manuscript and is shown in Supplementary Figure 1.

- In the histopathological comparison, was morphometry employed? How were conclusions regarding the inflammatory infiltrate drawn? Were there observable differences in other morphological changes between control and APRIL-tg groups? If so, these should be presented with corresponding morphometry 

The figures are only illustrative, no morphometry was used to comparatively observe the inflammatory process in the joint cavity of the animals used in the experiments. The figure was changed and a new panel in 20X was inserted. The legend of the figure was modified to explain panels.

- The discrepancy in IgG titer is mostly attributed to the IgG1 subclass (0.974 (APRIL-Tg mice) vs 0.594 (ctrl mice)). However, the actual IgG1 titer shows a less significant difference (0.666 (APRIL-Tg mice) vs 0.516 (ctrl mice). How do authors explain this? The reasons for this and the titers of other subclasses should be addressed. 

We thank the reviewer for this suggestion. In fact, we were based on the results by Stein J et al., 2002 (ref.[21] cited in the manuscript), showing an increase in IgM, but no difference in IgG and IgG1 in APRIL-Tg mice, and also on the data by Fernandez L et al., 2013 (ref. [18] cited in the manuscript) evaluating CIA on the model of APRIL-Tg DBA/1 mice, in which IgG1 and IgG2a are increased. So, we decided to investigate this issue in the context of our model of AIA in C57BL/6 APRIL-Tg mice. Furthermore, these immunoglobulins are cited as autoantibodies in humans with rheumatoid arthritis (Lewis BJB, et al., 2019, BMC Immunol. 2:44. doi: 10.1186/s12865-019-0328-6 and Miyata M, et. al., 1999, J Rheumatol. 26:1436-1438). Nevertheless, we only evaluated the IgG 1 subclass of IgG. Therefore, we agree that the IgG 1 subclass alone may not have been the only responsible for the difference in total IgG, and additional experiments are needed for evaluation of other IgG subclasses. This statement was included in the “Discussion” section of the manuscript. 

- The study's conclusion lacks clarity regarding the precise role of APRIL-driven mechanisms linked to B cell behavior in arthritis. It is essential to elucidate why B cells, in particular, are the focus over other cell types in the progression of arthritis in the context of APRIL's effects (in fact APRIL affects many cell types). Additionally, the rationale for emphasizing B cells should be made explicit, considering their role in comparison to other immune cells within the APRIL-mediated response in arthritis

The reason for choosing B cells as the focus of our study was that it is related to the induction of autoimmune rheumatoid arthritis, in a transgenic model for the inflammatory cytokine, APRIL, which over the years has been shown to modulate mainly B cells (Stein JV et al., 2002, ref.[21] cited in the manuscript; Fernandez L et al., 2013, ref. [18] cited in the manuscript; Planelles L et al., 2004. Cancer Cell. 6: 399-408 doi:10.1016/j.ccr.2004.08.033). Moreover, rheumatoid synovium is a site of significant B-cell infiltration, expansion, and differentiation (Townsend MJ et al., 2010, ref. [4] cited in the manuscript). In fact, B cells are related to autoimmune arthritis, our results demonstrated that even in Tg-APRIL the levels of autoantibodies remained high. Also, we observed that the expression of CXCL13 was increased in APRIL-Tg mice. 

Minor Concerns: 

- All plots should demonstrate individual points 

We opted for using column bar graphs representing mean ± SEM of each group because we were comparing homogeneous groups of animals (same sex, same age, defined APRIL-Tg mice and their control littermates), evaluating the variables control x arthritis. 

- In Figure 2, cellularity in lymph nodes and spleen is compared, but the methodology is unclear. If flow cytometry was used, how was it normalized? This information is needed 

As suggested, the methodology used for evaluation of cellularity in lymph nodes and spleen was added to the “Material and Methods” section of the manuscript. Cell counting was carried out in a Neubauer chamber at a dilution of 1:50, with trypan blue 10:10 v/v added. 

- The study should address whether there is a gender dependency in APRIL expression and justify the exclusive use of male mice 

Previous studies from our group showed that there is no difference considering the gender of animals in the expression of APRIL in transgenic mice (Stein JV et al., 2002 ref.[21] cited in the manuscript; Planelles et al., 2004 Cancer Cell. 6: 399-408. doi: 10.1016/j.ccr.2004.08.033; Fernandez L et al., 2013, Ref [18] cited in the manuscript). Considering this fact, and since during the present study a bigger number of males was born, we opted for using only male mice. 

- Details on single-cell suspension preparation should be added to the "Materials and Methods" section 

As suggested, details on single-cell suspension preparation were added to the “Materials and Methods” section. 

 The image (histological slides) acquisition technique needs description and inclusion in the "Materials and Methods" section (microscope, magnification etc.) 

As suggested, the information about microscope and magnification used to obtain histological images was included in “Material and Methods” section of the manuscript. 

- The current total of 11 figures could be consolidated into 5-6 for ease of reading, combining figures where feasible 

As suggested, the total of 11 figures was reduced, with consolidation of some figures, to a total of 8 at the final version of the manuscript. 

- Figure 3 requires redesigning for clarity, particularly in labeling images A and D, which currently reference healthy mice but appear to indicate acute and chronic arthritis states in the figure

We thank the reviewer for this suggestion. The ancient Figure 3 was included as part of Figure 1 in the revised manuscript. The figure was redesigned for clarity, showing (H) healthy littermate mouse, (i) healthy APRIL-Tg mouse, (J) AIA 3 days in littermate (acute), (K) AIA 3 days in April-Tg (acute), (L) AIA 10 days in littermate (chronic) and (M) AIA 10 days in APRIL-Tg (chronic).

---

## [Editor Report · Decision Letter 1]

3 Apr 2024

Anti-inflammatory role of APRIL by modulating regulatory B cells in antigen-induced arthritis

PONE-D-23-29381R1

Dear Dr. Villa-Verde,

We’re pleased to inform you that your manuscript has been judged scientifically suitable for publication and will be formally accepted for publication once it meets all outstanding technical requirements.

Kind regards,

Sadiq Umar

Academic Editor

PLOS ONE